# Using the VERT wearable device to monitor jumping loads in elite volleyball athletes

**Faraz Damji**[1,2], **Kerry MacDonald**[3,4], **Michael A. Hunt**[2], **Jack Taunton**[5], **Alex Scott**[1,2]*

**1** Centre for Hip Health and Mobility, Vancouver Coastal Health Research Institute, Vancouver, British Columbia, Canada, **2** Department of Physical Therapy, Faculty of Medicine, University of British Columbia, Vancouver, British Columbia, Canada, **3** Department of Athletics and Recreation, University of British Columbia, Vancouver, British Columbia, Canada, **4** School of Kinesiology, Faculty of Education, University of British Columbia, Vancouver, British Columbia, Canada, **5** Department of Family Medicine, Faculty of Medicine, University of British Columbia, Vancouver, British Columbia, Canada

* ascott@mail.ubc.ca

**Data Availability Statement:** All relevant data are within the manuscript and its Supporting Information files.

**Funding:** F.D. was the recipient of a Canadian Institute of Health Research Canada Graduate

## Abstract

Sport is becoming increasingly competitive and athletes are being exposed to greater physical demands, leaving them prone to injuries. Monitoring athletes with the use of wearable technology could provide a way to potentially manage training and competition loads and reduce injuries. One such technology is the VERT inertial measurement unit, a commercially available discrete wearable device containing a 3-axis accelerometer, 3-axis gyroscope and 3-axis magnetometer. Some of the main measurement outputs include jump count, jump height and landing impacts. While several studies have examined the accuracy of the VERT's measures of jump height and jump count, landing impact force has not yet been investigated. The objective of this research study was to explore the validity of the VERT landing impact values. We hypothesized that the absolute peak VERT acceleration values during a jump-land cycle would fall within 10% of the peak acceleration values derived simultaneously from a research-grade accelerometer (Shimmer). Fourteen elite university-level volleyball players each performed 10 jumps while wearing both devices simultaneously. The results showed that VERT peak accelerations were variable (limits of agreement of -84.13% and 52.37%) and had a propensity to be lower (mean bias of -15.88%) when compared to the Shimmer. In conclusion, the validity of the VERT device's landing impact values are generally poor, when compared to the Shimmer.

## Introduction

Athletes today are being exposed to high training loads and saturated competition calendars, due to the evolving nature of sport into a more competitive and professionalized industry. This creates an environment that is conducive to sports injuries. For example, tendinopathy is an overuse injury to the tendon and its insertions, and is associated with chronic tendon pain [1]. Those with tendinopathy experience a swollen, painful tendon caught in a cumulative cycle of injury-repair which can last months or years [2]. Typically, these people are advised to scale back the intensity of their sports participation while they participate in rehabilitation, and then gradually increase tendon loading activities whilst monitoring and minimizing pain [3].

Scholarship - Master's Award, which provided funding for this study (CIHR CGS-M, http://www.cihr.ca/). A.S. was a recipient of a Natural Sciences and Engineering Research Council of Canada Discovery Grant, which also provided funding for this study (NSERC DG, https://www.nserc-crsng.gc.ca/). The funders had no role in study design, data collection and analysis, decision to publish, or preparation of the manuscript.

**Competing interests:** The authors have declared that no competing interests exist.

Evidence supports that poor load management is a major risk factor for injury [4]. In the context of sports medicine, load can be defined broadly as "the sport and non-sport burden (single or multiple physiological, psychological or mechanical stressors) as a stimulus that is applied to a human biological system (including subcellular elements, a single cell, tissues, one or multiple organ systems, or the individual)" [4]. Additionally, it may be applied "over varying time periods (seconds, minutes, hours to days, weeks, months and years) and with varying magnitude (ie, duration, frequency and intensity)" [4].

Monitoring training and competition load placed on athletes is gaining popularity as a fundamental practice to ensure they are being subjected to appropriate and therapeutic levels [5]. A new generation of wearable technologies are available, which enable the precise quantification of load with built-in accelerometers, gyroscopes and magnetometers. One such technology is the VERT system (www.myvert.com), a commercially available discrete wearable device that measures vertical displacement through a proprietary algorithm [6]. This inertial measurement unit (IMU) contains a 3-axis accelerometer, 3-axis gyroscope and 3-axis magnetometer. The dimensions of the unit are 6 x 3 x 0.5 cm and it is typically worn on an elastic belt at the level of the L3 or L4 vertebrae, thought to be near the body's center of mass. The basic model records jump count and jump height and a newer G-VERT model also records landing impact and kinetic energy.

Several studies have examined the accuracy of the VERT's measures of jump height and jump count, and generally shown it to be acceptable [6–13]. Landing impact, however, is a key parameter that has not yet been investigated. In sport, the act of landing from a jump is often required and may happen very frequently [14]. The resulting ground reaction forces on the body during landing have been shown to be influential in lower limb injury [15]. It is apparent that the movement of joints and muscle activity in the lower limb can decrease the magnitude of these impact forces. The VERT calculates landing impact as the instantaneous acceleration values resulting from forces upon landing, expressed as a G-force ($1G = 9.81 m/s^2$). A resultant value is given, encompassing all components of acceleration in the X-Y-Z axes. This study was the first to perform an external validation of VERT for the landing impact parameter.

The objective of this research study was to examine the accuracy of VERT landing impact values in university volleyball players. We hypothesized that the absolute VERT landing impact values would fall within 10% of the landing impact values derived from a research-grade accelerometer. The specific research-grade accelerometer that was chosen for this study has been previously used extensively to monitor human health and track activities of daily living [16–19]. Information about the design, sensing capabilities and firmware are available to the general public [20]. The acceleration signals from the device, when compared with force platform and motion capture data, were shown to be valid in terms of the acceleration-based measurements for spatio-temporal gait parameters [21,22].

## Methods

This study was approved by the Clinical Research Ethics Board at the University of British Columbia (Certificate Number: H19-00163). Each participant provided written informed consent.

### Participants

The study population included 14 varsity volleyball players at the University of British Columbia (UBC), with 11 players recruited from the men's volleyball team and 3 players recruited from the women's volleyball team. Our initial plan was to recruit equal numbers of men and women, but data collection was terminated due to the COVID-19 pandemic shortly after the

start of data collection with the women's volleyball team. We excluded participants who had been instructed to avoid jumping (e.g. due to injury). Participants filled out a modified version of the Oslo Sport Trauma Research Centre-Patellar Tendinopathy (OSTRC-P) questionnaire which has been shown to be valid for self-reporting patellar tendinopathy symptom severity in youth basketball [23]. Body mass and height measurements were recorded with a portable weigh scale (Weight Watchers 14C, Conair Consumer Products, Woodbridge, Ontario) and standing height measure (HM200P, Charder Medical, Taichung, Taiwan). Standing reach height was measured with the Vertec (5013, Jump USA, Sunnyvale, California). Ethical approval from the Clinical Research Ethics Board at UBC was obtained (Certificate Number: H19-00163). All participants provided informed consent.

### Study design

A single research-grade accelerometer (Shimmer3 IMU, Shimmer Sensing, Dublin, Ireland) was used as the comparator for the VERT, and both devices were worn adjacently with one device placed to the right of the navel and the other to the left of the navel. This allowed for sufficient spacing to prevent contact between devices. The VERT was held securely in the commercially available VERT elastic waistband, in the same position and orientation intended by the developers. The Shimmer was clipped into the Shimmer adjustable belt, which was worn directly on top of the VERT waistband. The Shimmer was calibrated using the Shimmer 9DOF Calibration software, and the wide range accelerometer data output was selected with a range of +/- 16g, and sampling frequency of 2048 Hz.

Data collection occurred at the university volleyball court, a hardwood surface. After an optional warm-up at a self-selected pace on a stationary bike and 2–3 practice jumps, each participant performed a total of 10 countermovement jumps (CMJs) while collecting data simultaneously from both the VERT and Shimmer devices. The first block consisted of 5 maximal CMJs. The second block consisted of 5 submaximal CMJs at 80% of the maximum height achieved in the first block, measured using a Vertec. The rest periods were 20 seconds between trials of the same block and 2 minutes between blocks, to avoid the effects of fatigue and allow for clear data visualization.

### VERT and Shimmer data processing

During the study visit, both the highest magnitude impact and VERT-identified landing impact values were recorded for each jump, as displayed on the VERT iPad application (Fig 1).

As seen in Fig 1 (a screenshot from the VERT iPad application), the *Impacts Vs. Time graph* displays the resultant accelerations for each jump/land cycle. Yellow bars denote VERT-identified landing impact values, and red bars denote accelerations categorized by the VERT software to be from accelerations other than landing impact. The VERT excel file downloaded from the myVERT online server shows the same impact data with a higher degree of precision and time-stamped. For each jump, the highest magnitude acceleration and VERT-identified

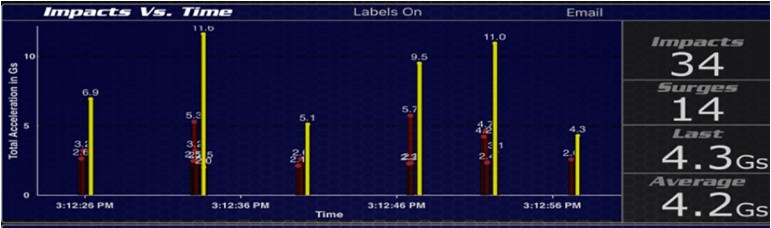

**Fig 1. Screenshot from VERT iPad application.**

landing impact values were located in the excel file and the exact values were used to replace the initially documented values.

The Shimmer Excel file was exported from the ConsensysPRO software into Excel, yielding a series of time-stamped acceleration values for each of the X-Y-Z axes in m/s^2. The resultant accelerations were calculated from the separate components using the formula:

$$Resultant\ acceleration = \sqrt{[(acceleration\ in\ X)^2 + (acceleration\ in\ Y)^2 + (acceleration\ in\ Z)^2]}$$

These values were then converted from m/s^2 to Gs. Scatter plots were generated with spikes corresponding to jumps clearly visible (Fig 2). The maximum values of each spike, or the resultant peak accelerations, could then be directly compared to values from the VERT.

## Statistical analysis

For each pair of measurements from each jump, the percentage difference between Shimmer and VERT peak accelerations were calculated. A Bland-Altman plot approach was then used to assess the discrepancy between devices across the range of G values, with adjustments made due to the presence of repeated measures [24,25]. Intraclass correlation coefficients (ICC) and concordance correlation coefficients (CCC) were also calculated. There are many versions of CCC, and in this analysis we used the version which is computed from variance components of a linear mixed effect model for longitudinal repeated measures [26]. We additionally assumed a compound symmetric covariance structure.

## Results

### Participant demographics

The demographics for the 14 participants can be found in Table 1.

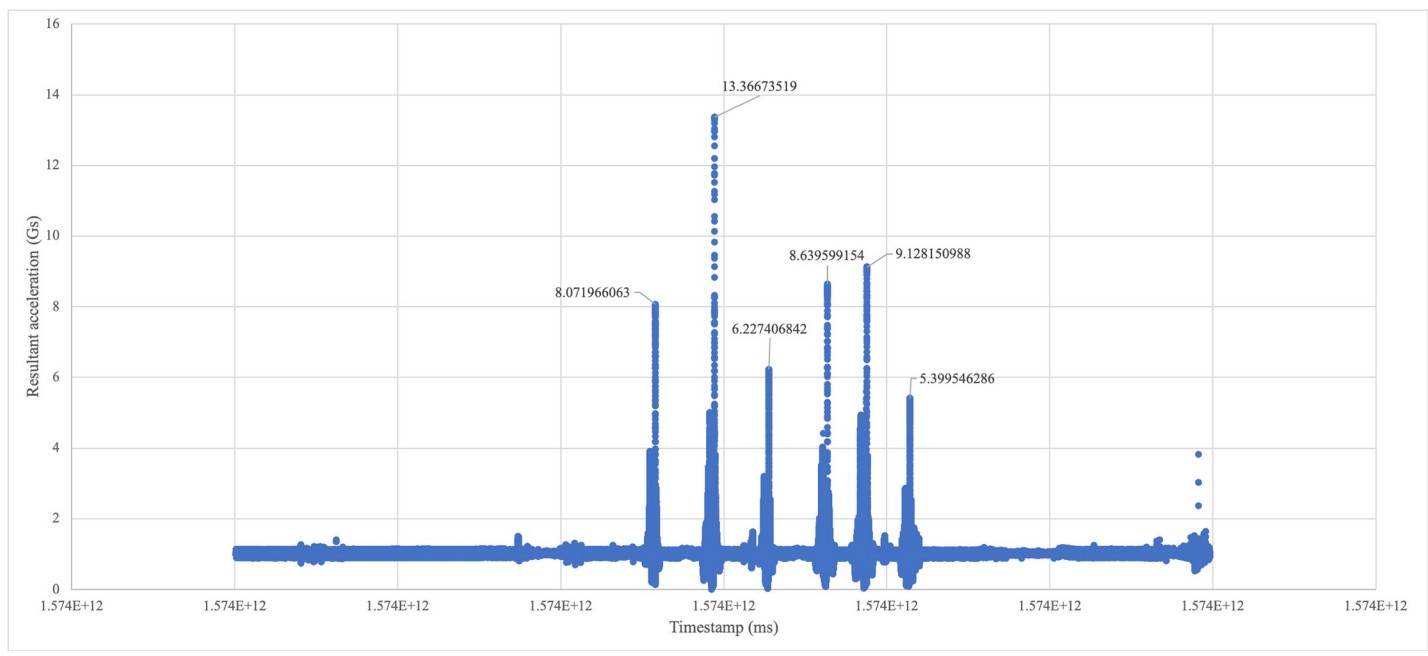

**Fig 2. Shimmer scatter plot generated from Shimmer excel data.**

**Table 1. Participant demographics.**

| Variable | All Subjects (n = 14) |
|---|---|
| Age (yr), mean ± SD | 20.9 ± 1.5 |
| Male | 11 |
| Female | 3 |
| Height (cm), mean ± SD | 187.6 ± 10.5 |
| Standing reach (cm), mean ± SD | 244.6 ± 15.2 |
| Body mass (kg), mean ± SD | 80.9 ± 10.8 |
| Maximum jump height (cm), mean ± SD | 59.2 ± 10.2 |
| Self-reported patellar tendinopathy in one or both knees on OSTRC-P [a] | 3 |

SD = standard deviation.

[a] OSTRC-P data was missing for 2 of the 14 participants, who accidentally missed the form during their study visits.

## Comparison of peak accelerations between VERT and Shimmer

Summary statistics of the devices followed a similar trend. We observed that the mean VERT value for peak acceleration (7.8 G) was lower than the Shimmer (10.0 G). The standard deviations of the VERT and Shimmer were similar (3.5 G and 4.1 G, respectively), as well as the maximal (24.3 G and 21.2 G, respectively) and minimal (3.6 G and 4.1 G, respectively) values for each device.

The results of the analysis are shown in the Bland-Altman plot below (Fig 3).

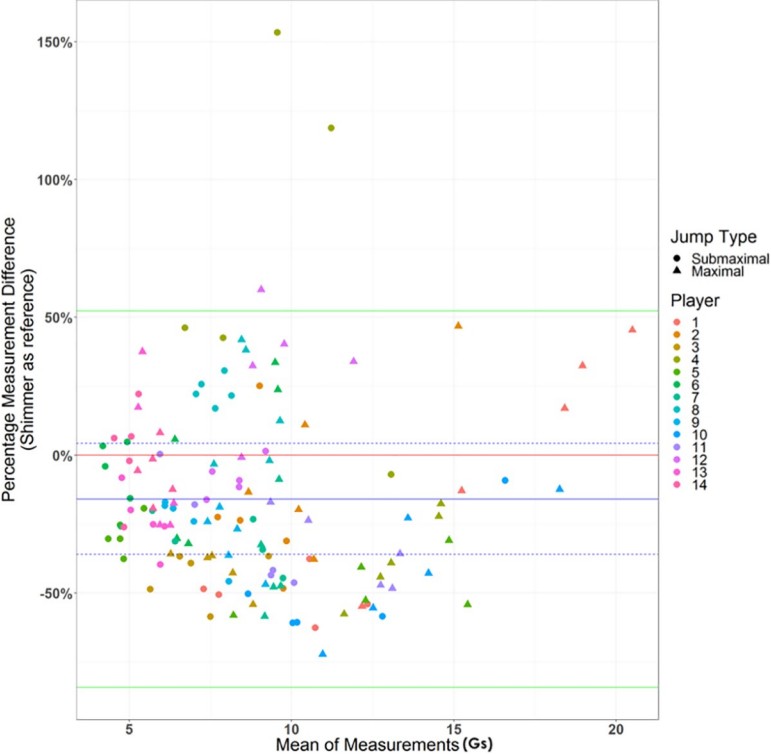

**Fig 3. Bland-Altman plot for VERT and Shimmer agreement.** The red line represents the 0% difference mark. The blue line represents the mean bias, or the mean percentage difference we observe between VERT and Shimmer. The dotted lines around the mean bias represent the 95% confidence interval. Lastly, the green lines represent limits of agreement, which are the (mean bias)+1.96*(standard deviation of differences) and (mean bias)-1.96*(standard deviation of differences).

The limits of agreement within this plot are remarkably wide, with limits of -84.13% and 52.37%, suggesting that percentage differences between VERT and Shimmer vary by a large amount. Additionally, when examining the mean bias, we observe that its value is -15.88%. The confidence interval spans from -35.99% to 4.23%. As it overlaps with 0%, it suggests the mean bias may not be significantly different from 0%, but this is likely attributed to the high variability that we observe in the differences. With such a wide-ranging confidence interval, the mean bias may nevertheless be large and negative since the lower bound of the confidence interval stretches down to -35.99%. There may be a tendency for VERT to have lower measurements than Shimmer, resulting in a bias.

In examining the distribution of points themselves, it is evident that when the mean between the pairs of measurements is less than 10, then the points are nearly evenly distributed around the mean bias. When the mean of a measurement pair exceeds 10, then a peculiar weak trend is seen where percentage differences are very low but increase towards the mean bias and beyond. This is indicative of lower reliability, and hence validity, of the VERT device at landing impacts on the higher end of the spectrum.

The ICC (two-way mixed effect model, single measures, and absolute agreement) calculated is 0.49 (95% CI: 0.38 to 0.60) [27]. The moderate size of the ICC value indicates some agreement between the two devices, but agreement is generally mediocre. The calculated CCC is 0.37 (95% CI: 0.12 to 0.58). The mild size of the CCC value corroborates what we observe with the ICC value, although the measure of agreement is worse here once we account for the repeated measures within subjects.

## Discussion

### Main findings

The aim of this study was to examine the accuracy of VERT landing impact values in university volleyball players. It was hypothesized that the VERT would generate peak acceleration values similar to the Shimmer. Contrary to our hypothesis, the VERT peak acceleration values were highly variable, as evidenced by the percentage differences with Shimmer as a reference, and there is a propensity for VERT to produce lower measurements than Shimmer. Hence, the validity of the VERT device's landing impact values can be said to be generally poor, with respect to Shimmer, when it comes to measuring landing impacts. This is contrary to our hypothesis that the absolute VERT landing impact values will fall within 10% of the landing impact values derived from the Shimmer.

As this study was the first to perform an external validation of VERT for the landing impact parameter, the main findings cannot be discussed in relation to previous literature but several considerations could explain these findings. First, it is possible that the sampling rate of the VERT is not sufficiently high to meet the performance demands of volleyball. Second, the disparate nature of the data from each of the devices can also serve as an explanation for our results. While the VERT processes the data through a proprietary algorithm which was not available to us, the Shimmer provides the full signal. Another consideration is that landing biomechanics such as the amount of hip or knee flexion, or placement of the feet, likely differed both within and between individuals. This variability may have led to different acceleration values at the exact location of the VERT and Shimmer sensors. This possibility could be tested by placing two Shimmers side by side in the same positions as used for this study, and assessing the magnitude of difference.

Some discussion is warranted around the issue with the VERT-identified landing impact values. The VERT identifies a number of acceleration peaks for each jump-land cycle. This generally is seen as a cluster of vertical bars on the measurement output produced by the

VERT software. For the most part, each cluster consists of one yellow bar (VERT-identified landing impact) and the rest are all red bars (accelerations due to events other than landing). The yellow bar is the tallest in some instances, but this is not always the case. Neither does there seem to be any temporal pattern; in other words, the order of the red and yellow bars seems to be random for each cluster. The VERT also may (incorrectly) show multiple yellow bars in a given cluster because of heel to toe landings, one foot hitting the ground before the other foot, or simply a hard landing with a subtle reverberation. For these reasons, we chose to use the single highest acceleration peak for each cluster that appeared in the VERT data. Shimmer data was similar to the VERT data in that it also showed a number of acceleration peaks for each jump-land cycle. We made the assumption that the highest magnitude acceleration from a jump-land cycle corresponds to landing impact. It is true that this may not always be the case, however an argument can be made that our assumption stands the majority of the time. A study that examined the application of force during the vertical jump, in both children and adults, shows force-time curves that support our reasoning [28].

Although the VERT's sampling rate was unknown, we had good reason to believe that the Shimmer sampling rate of 2048 Hz was significantly higher than the VERT. Many other studies with similar methodology have ensured equal sampling rates in devices being used. Our rationale for using the highest possible sampling rate as a control was to allow us to examine the possibility that the VERT may be underestimating landing impact values, which would occur if sampling frequency is too low to capture the peak values.

## Limitations

Force platforms, the gold standard for measuring forces, were not used in this study for validating the VERT and this presents a limitation. Research-grade accelerometers, such as the Shimmer device, were a suitable and possibly better alternative for a few reasons. Firstly, the Shimmer presents raw data and does not use proprietary algorithms unlike the VERT. Secondly, the Shimmer has been compared with force platform data. A study comparing force calculated from the Shimmer device with force platform data found moderate to low levels of agreement and a consistent systematic bias between both technologies [29]. Although this is not the ideal and desired outcome, this means the Shimmer could still be used as a proxy to infer how far off from the gold standard the VERT stands. In a study by Howard et al. (2014), the Shimmer generally overestimated peak forces compared to force platform data. If the Shimmer is overestimating, and the VERT is estimating lower than the Shimmer, the VERT may either be overestimating or underestimating compared to the gold standard force platform. A future study could be conducted to evaluate how the VERT performs in relation to the force platform. However, it should be noted that a force platform will be only partly related to the resultant peak acceleration at the worn location of the VERT due to the dissipation of force travelling up the kinetic chain.

While evidence-based guidelines were being followed to mitigate any potential interference between the two devices (e.g. avoiding placing one on top of the other), there was a chance that there was crosstalk when in close proximity at the level of the waist [30]. Despite taking the steps to ensure the waist bands were fit snug and positioning the devices far apart to avoid bumping into each other, unwanted movement of the bands or devices may have still occurred without knowing.

## Practical recommendations

Based on the findings of this study, coaches and practitioners should treat the VERT landing impact data cautiously. This certainly does not mean that the VERT is not useful, as it has proven to be valid and reliable for measuring jump height and jump count.

## Conclusions

In conclusion, the VERT landing impact values were more variable and had a propensity to be lower when compared to the same measurements taken from a research-grade accelerometer. Hence, the validity of the VERT device's landing impact values can be said to be generally poor, with respect to the Shimmer. While the VERT may be a useful tool for the practice of load monitoring, the landing impact data may not be truthful and caution should be exercised with the interpretation of this parameter until further validation studies are conducted.

## Supporting information

**S1 File. Full de-identified dataset.**
(XLSX)

## Acknowledgments

We would like to acknowledge the efforts of Eric Sanders, Nikolas Krstic and Johann Windt for assisting with the statistical analysis, and David Gil for his expert advice on the VERT wearable technology.

## Author Contributions

**Conceptualization:** Faraz Damji, Kerry MacDonald, Michael A. Hunt, Alex Scott.

**Data curation:** Faraz Damji, Alex Scott.

**Formal analysis:** Faraz Damji, Alex Scott.

**Investigation:** Faraz Damji, Alex Scott.

**Methodology:** Faraz Damji, Kerry MacDonald, Michael A. Hunt, Jack Taunton, Alex Scott.

**Project administration:** Faraz Damji, Kerry MacDonald, Michael A. Hunt, Jack Taunton, Alex Scott.

**Supervision:** Kerry MacDonald, Michael A. Hunt, Jack Taunton, Alex Scott.

**Validation:** Faraz Damji, Kerry MacDonald, Michael A. Hunt, Jack Taunton, Alex Scott.

**Visualization:** Faraz Damji, Kerry MacDonald, Michael A. Hunt, Jack Taunton, Alex Scott.

**Writing – original draft:** Faraz Damji, Alex Scott.

**Writing – review & editing:** Faraz Damji, Kerry MacDonald, Michael A. Hunt, Jack Taunton, Alex Scott.

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
