## [Decision Letter · Decision Letter 0]

4 Sep 2020

PONE-D-20-21022

Using the VERT wearable device to monitor jumping loads in elite volleyball athletes

PLOS ONE

Dear Dr. Scott,

Thank you for submitting your manuscript to PLOS ONE. After careful consideration, we feel that it has merit but does not fully meet PLOS ONE’s publication criteria as it currently stands. Therefore, we invite you to submit a revised version of the manuscript that addresses the points raised during the review process.

We look forward to receiving your revised manuscript.

Kind regards,

José M. Muyor

Academic Editor

PLOS ONE

Reviewers´ comments

Reviewer 1

The study deals with a comparison between two IMUs (with different sensor specs) during volleyball jumping. The authors state that the consumer grade VERT device would provide similar (within a 10% range) peak acceleration compared to a “research grade” sensor.

While the idea of the study is nice it must be stated that the theoretical background does not directly lead to the research question. The VERT device does not provide any sensor specifications which already is a red flag. While the system provides jump count metrics the idea of the authors was to compare peak acceleration between the two sensors. In my opinion a simulation study could be conducted first to back up the idea that a sensor with unknown sensor specs will still provide similar results as the “gold standard” which should have been the force plate. Using the Shimmer sensor device (+/- 16g, and sampling frequency of 2048 Hz) during a jump could give some insights of how various sensor specs may affect the peak acceleration. The comparison between the two devices is also a little flawed as it is not clear if the peak acceleration actually occurred during the landing period, which could have been tested with a force plate. I understand not every laboratory has a force plate but in this case I would have started out with a technical comparison. Put both accelerometers on a sled or on a S&C machine to keep the same movement going (or barbell etc) with an a similar fixation (Latissiums machine and fix them next to each other on the weights).

The problem with human testing is always the sensor placement. Has the sensor moved? Were the sensor positions changed? How is it guaranteed that the sensors actually do measure the same and one not its own movement also? Why not put both sensors on a rigid object and fix it to the body?

Overall I like your approach but I suggest you go back to the drawing board and get some technical evidence before you run the jumping experiment. Fourteen subjects is not a lot and while I usually refrain from telling authors to redo an experiment, I think that once COVID-19 has been conquered you could get more people in the lab. Also you don’t need elite athletes, as you look at jumping in general and your tests were not volleyball or high performance related. A CMJ does not reflect real playing behavior…

Please use the Equation tool for your signal magnitude calculation (also not really necessary as this is a very easy and standard procedure in signal processing)

Table 2 could be in the text so please remove

If you want to test the accuracy of VERT landing impact then you need to compare it to the Gold Standard but you also state that Shimmer is not the gold standard…

Reviewer 2

General comments: The authors describe a descriptive research study that compared the simultaneous measurement of impact acceleration measures during jump landings in collegiate volleyball players. A commercially available device marketed to sports users (Vert) was compared to a commercially available research-grade accelerometer (Shimmer). There was poor agreement and concordance demonstrated for peak impact acceleration measures between the two devices. The authors identify several potential causes for the results but the speculative nature of these reasons lead to questions about the methodology of the current study. These questions cast doubt on the internal validity of the current study methods.

Specific comments:

Lines 54-58: The sentences describing the pathophysiology of tendonopathy seem out of place in the introduction. Recommend deleting.

Lines 81-84: Are there references to support the benefits and limitations of Vert? Or, are these the authors’ observations? If the latter, other than the lack of accuracy studies I don’t think that this content fits in the introduction.

Lines 102-108: I’m curious as to the choice of a research-grade accelerometer (Shimmer) as the gold standard for this validation study. Many sports biomechanists would consider a force plate to be the gold standard for measuring landing impacts. Citing studies that report the validity of the Shimmer measures compared to force plate measures during walking gait are helpful but jump landings occur at much higher speed than walking so the relevance of the walking studies are marginal in relation to the current study. I believe that further justification of using the Shimmer as the gold standard measurement device in this study is warranted.

Lines 114-115: Was an a priori sample size estimate performed?

Lines 132-133: The placement of the two devices isn’t clear. Was one device placed to the right of the navel and the other to the left of the navel? Or were both devices placed on either the left or the right side? Please revise to improve clarity.

Line 135: What was the sampling rate for Vert?

Line 137: More information on the court surface would be helpful. Was it hardwood or a synthetic surface?

Line 176: Spelling should be “Altman”

Lines 206-210: This information seems better suited as the figure legend rather than in the text of the Results section.

Line 229: I recommend deleting this sentence as it is not a statistically-based result.

Line 255: Report the sampling rate for Vert if that information is publicly available.

Lines 257-258: Expand on the potential difference in algorithms for data processing between the two devices.

Lines 258-260: I fail to see how variability in jump technique would influence the lack of agreement between the two measurement devices. Are you suggesting that there was a right-to-left asymmetry in jumping mechanics that would have affected the measurements by the sensors placed on the right or left side of the body?

Lines 261-263: Why wasn’t this done to ensure that sensor placement didn’t influence the primary results of the study. This seems like a glaring flaw that limits reader confidence in the current results.

Line 282: Were efforts made to contact the manufacturer of Vert to ascertain the sampling rate? If yes, explicitly state this. If no, I strongly encourage the authors to do so.

Lines 295-299: The reasoning justifying the use of Shimmer as the gold standard (in lieu of a force plate) seems flawed based on the results of reference #30.

Lines 311: This is the first mention of “waist bands”. Please elaborate on this in the methods section.

Lines 315-353: This section does not contextualize the results of the current investigation and seems disconnected from the rest of the manuscript.

References: None of the journal articles references have volume or page numbers listed.(less...)

Reviewers' comments:

Reviewer's Responses to Questions

**Comments to the Author**

1. Is the manuscript technically sound, and do the data support the conclusions?

Reviewer #1: Yes

Reviewer #2: Partly

2. Has the statistical analysis been performed appropriately and rigorously? 

Reviewer #1: Yes

Reviewer #2: Yes

3. Have the authors made all data underlying the findings in their manuscript fully available?

Reviewer #1: Yes

Reviewer #2: Yes

4. Is the manuscript presented in an intelligible fashion and written in standard English?

Reviewer #1: Yes

Reviewer #2: Yes

6. PLOS authors have the option to publish the peer review history of their article (what does this mean?). If published, this will include your full peer review and any attached files.

Reviewer #1: No

Reviewer #2: No

---

## [Author Response · Author response to Decision Letter 0]

2 Nov 2020

Reviewers´ comments

Reviewer 1

The study deals with a comparison between two IMUs (with different sensor specs) during volleyball jumping. The authors state that the consumer grade VERT device would provide similar (within a 10% range) peak acceleration compared to a “research grade” sensor.

While the idea of the study is nice it must be stated that the theoretical background does not directly lead to the research question. The VERT device does not provide any sensor specifications which already is a red flag. While the system provides jump count metrics the idea of the authors was to compare peak acceleration between the two sensors. In my opinion a simulation study could be conducted first to back up the idea that a sensor with unknown sensor specs will still provide similar results as the “gold standard” which should have been the force plate. Using the Shimmer sensor device (+/- 16g, and sampling frequency of 2048 Hz) during a jump could give some insights of how various sensor specs may affect the peak acceleration. The comparison between the two devices is also a little flawed as it is not clear if the peak acceleration actually occurred during the landing period, which could have been tested with a force plate. I understand not every laboratory has a force plate but in this case I would have started out with a technical comparison. Put both accelerometers on a sled or on a S&C machine to keep the same movement going (or barbell etc) with an a similar fixation (Latissiums machine and fix them next to each other on the weights).

The problem with human testing is always the sensor placement. Has the sensor moved? Were the sensor positions changed? How is it guaranteed that the sensors actually do measure the same and one not its own movement also? Why not put both sensors on a rigid object and fix it to the body?

Overall I like your approach but I suggest you go back to the drawing board and get some technical evidence before you run the jumping experiment. Fourteen subjects is not a lot and while I usually refrain from telling authors to redo an experiment, I think that once COVID-19 has been conquered you could get more people in the lab. Also you don’t need elite athletes, as you look at jumping in general and your tests were not volleyball or high performance related. A CMJ does not reflect real playing behavior…

Please use the Equation tool for your signal magnitude calculation (also not really necessary as this is a very easy and standard procedure in signal processing)

Table 2 could be in the text so please remove

If you want to test the accuracy of VERT landing impact then you need to compare it to the Gold Standard but you also state that Shimmer is not the gold standard…

Thank you for your detailed and constructive feedback. While your queries about the VERT sensor specifications have been answered in the responses to the specific comments below, I will address your other comments here. In our study, we were very much interested in testing the VERT while worn in the position and orientation intended by the developers. Although it could have been beneficial to place the accelerometers on a machine or fixation, this defeated our purpose of conducting a very practical validation study that was as close to real-life conditions as possible. We were willing to accept the potential problems (such as chances of the sensors moving) for this reason. The reason we opted for CMJs as opposed to volleyball specific jumps was to encourage landing on two feet and in turn, data that was easier to interpret/analyze.

Reviewer 2

General comments: The authors describe a descriptive research study that compared the simultaneous measurement of impact acceleration measures during jump landings in collegiate volleyball players. A commercially available device marketed to sports users (Vert) was compared to a commercially available research-grade accelerometer (Shimmer). There was poor agreement and concordance demonstrated for peak impact acceleration measures between the two devices. The authors identify several potential causes for the results but the speculative nature of these reasons lead to questions about the methodology of the current study. These questions cast doubt on the internal validity of the current study methods.

Thank you for your time spent reviewing our manuscript. Hopefully the responses to the specific comments below will provide further insights about the methodology of our study.

Specific comments:

Lines 54-58: The sentences describing the pathophysiology of tendonopathy seem out of place in the introduction. Recommend deleting.

The details of the pathophysiology have been removed. 

Lines 81-84: Are there references to support the benefits and limitations of Vert? Or, are these the authors’ observations? If the latter, other than the lack of accuracy studies I don’t think that this content fits in the introduction.

These are the authors observations, therefore no references were included. The discussion on the benefits and limitations has now been removed.

Lines 102-108: I’m curious as to the choice of a research-grade accelerometer (Shimmer) as the gold standard for this validation study. Many sports biomechanists would consider a force plate to be the gold standard for measuring landing impacts. Citing studies that report the validity of the Shimmer measures compared to force plate measures during walking gait are helpful but jump landings occur at much higher speed than walking so the relevance of the walking studies are marginal in relation to the current study. I believe that further justification of using the Shimmer as the gold standard measurement device in this study is warranted.

The rationale for selecting the Shimmer was two-fold. Firstly, the Shimmers were readily available to us whereas we did not have access to portable force platforms in the biomechanics lab. Secondly, the Shimmers have been used extensively in the literature (including validation studies) and information about their technical specifications is open to the public. In addition, there are possibly some reasons why the Shimmer would be a better alternative to force platforms as discussed in the limitations section of this paper. For these reasons, we decided that the Shimmer is the next best option to the force platform.

The aim of this study was to provide a first look at the validity of the VERT landing impact values, a novel contribution. We expect to see future research evaluating the VERT in relation to the force platform, and realize this is needed to be more certain of the validity. 

Lines 114-115: Was an a priori sample size estimate performed?

It was deemed that no sample size estimate was necessary for this particular study, given its exploratory nature. The sample size was determined based on the study designs of other validations of the VERT that have been conducted in the past (analyzing jump height and jump count). It is also important to note that the number of participants was limited by the COVID crisis which led to an early termination of our data collection.

Lines 132-133: The placement of the two devices isn’t clear. Was one device placed to the right of the navel and the other to the left of the navel? Or were both devices placed on either the left or the right side? Please revise to improve clarity.

Apologies for the lack of clarity on the placement of the devices. The language has been revised.

Line 135: What was the sampling rate for Vert?

The sampling rate for the VERT is proprietary information, and is not publicly available. For the purpose of this study, the VERT performance lab director privately disclosed this information to us, but we do not feel it is appropriate to include the actual sampling rate in the text. In all of the existing VERT validation studies, this proprietary information is not disclosed.

Line 137: More information on the court surface would be helpful. Was it hardwood or a synthetic surface?

The flooring of the university volleyball court was hardwood. This information has been included. 

Line 176: Spelling should be “Altman”

Thank you for the correction.

Lines 206-210: This information seems better suited as the figure legend rather than in the text of the Results section.

Thank you, this has been amended.

Line 229: I recommend deleting this sentence as it is not a statistically-based result.

This sentence has been deleted.

Line 255: Report the sampling rate for Vert if that information is publicly available.

As previously mentioned, we cannot report this value unfortunately.

Lines 257-258: Expand on the potential difference in algorithms for data processing between the two devices.

We do not have any information regarding the VERT algorithms. On the other hand, the Shimmer data is raw data. That is, no algorithms were applied. 

Lines 258-260: I fail to see how variability in jump technique would influence the lack of agreement between the two measurement devices. Are you suggesting that there was a right-to-left asymmetry in jumping mechanics that would have affected the measurements by the sensors placed on the right or left side of the body?

Yes, you are correct. We are suggesting there could have been a right-to-left asymmetry in jumping mechanics. 

Lines 261-263: Why wasn’t this done to ensure that sensor placement didn’t influence the primary results of the study. This seems like a glaring flaw that limits reader confidence in the current results.

We were able to reduce the likelihood of asymmetrical landings (i.e. one foot before the other) by specifically instructing participants to reach with two hands when performing the CMJs. This ensured landing on two feet. Additionally, any landings that were visibly asymmetrical during the data collection were identified and participants were asked to repeat these attempts. Only the repeats were included in the analysis. 

Line 282: Were efforts made to contact the manufacturer of Vert to ascertain the sampling rate? If yes, explicitly state this. If no, I strongly encourage the authors to do so.

Kindly refer to the earlier responses regarding the VERT sampling rate.

Lines 295-299: The reasoning justifying the use of Shimmer as the gold standard (in lieu of a force plate) seems flawed based on the results of reference #30.

The intent was to acknowledge that although the results of reference #30 shows moderate-low agreement between the Shimmer and force plate, the Shimmer does have its benefits due to its nature as a discrete wearable device that can be positioned at different locations on the body.

Lines 311: This is the first mention of “waist bands”. Please elaborate on this in the methods section.

This has been clarified in the methods section.

Lines 315-353: This section does not contextualize the results of the current investigation and seems disconnected from the rest of the manuscript.

The future directions section has been deleted from the manuscript.

---

## [Decision Letter · Decision Letter 1]

2 Dec 2020

PONE-D-20-21022R1

Using the VERT wearable device to monitor jumping loads in elite volleyball athletes

PLOS ONE

Dear Dr. Scott,

Thank you for submitting your manuscript to PLOS ONE. After careful consideration, we feel that it has merit but does not fully meet PLOS ONE’s publication criteria as it currently stands. Therefore, we invite you to submit a revised version of the manuscript that addresses the points raised during the review process.

We look forward to receiving your revised manuscript.

Kind regards,

José M. Muyor

Academic Editor

PLOS ONE

Reviewers' comments:

Reviewer's Responses to Questions

**Comments to the Author**

1. If the authors have adequately addressed your comments raised in a previous round of review and you feel that this manuscript is now acceptable for publication, you may indicate that here to bypass the “Comments to the Author” section, enter your conflict of interest statement in the “Confidential to Editor” section, and submit your "Accept" recommendation.

Reviewer #1: All comments have been addressed

Reviewer #2: (No Response)

2. Is the manuscript technically sound, and do the data support the conclusions?

Reviewer #1: Partly

Reviewer #2: Yes

3. Has the statistical analysis been performed appropriately and rigorously? 

Reviewer #1: Yes

Reviewer #2: Yes

4. Have the authors made all data underlying the findings in their manuscript fully available?

Reviewer #1: Yes

Reviewer #2: Yes

5. Is the manuscript presented in an intelligible fashion and written in standard English?

Reviewer #1: Yes

Reviewer #2: Yes

6. Review Comments to the Author

Reviewer #1: Dear authors, during the initial review I contacted the VERT company and even received the sensor specifications so you could have been a little more "brave" in disucssing the sensor specifications. Having said this, I think you have improved the manuscript and it is up to standard. I wish you had improved your presentation of the results a little to make a stronger point for your case. In the end your conclusion is: Shimmer is better in detecting peak accelerations which is a little "simple". Not every coach is really interested in the peak accelerations as they clearly depend on the sensor location, and sensor specs. Anyways, sorry to hear that your data collection was cut short due to the current global pandemic. Good luck with your future research!

Reviewer #2: The authors have adequately addressed my major concerns from the initial submission and the revised manuscript represents a marked improvement. I have a few further minor comments:

Line 115: Participants’ scores on the OSTRC-P should be reported with along with the other demographic measures in Table 1.

Line 264: “on the ipad” – it would be better to say “on the measurement output produced by the Vert software”.

Line 271: Change “preferred” to “chose”

7. PLOS authors have the option to publish the peer review history of their article (what does this mean?). If published, this will include your full peer review and any attached files.

Reviewer #1: No

Reviewer #2: No

---

## [Author Response · Author response to Decision Letter 1]

19 Dec 2020

Reviewers' comments:

Reviewer #1: Dear authors, during the initial review I contacted the VERT company and even received the sensor specifications so you could have been a little more "brave" in disucssing the sensor specifications. Having said this, I think you have improved the manuscript and it is up to standard. I wish you had improved your presentation of the results a little to make a stronger point for your case. In the end your conclusion is: Shimmer is better in detecting peak accelerations which is a little "simple". Not every coach is really interested in the peak accelerations as they clearly depend on the sensor location, and sensor specs. Anyways, sorry to hear that your data collection was cut short due to the current global pandemic. Good luck with your future research!

We appreciate all of your feedback and attention to detail, which has improved the quality of our work. 

Reviewer #2: The authors have adequately addressed my major concerns from the initial submission and the revised manuscript represents a marked improvement. I have a few further minor comments:

Thank you very much. Please see the responses to your specific comments below.

Line 115: Participants’ scores on the OSTRC-P should be reported with along with the other demographic measures in Table 1.

OSTRC-P scores are now included in Table 1.

Line 264: “on the ipad” – it would be better to say “on the measurement output produced by the Vert software”.

This has been amended.

Line 271: Change “preferred” to “chose”

This has been amended.

---

## [Editor Report · Decision Letter 2]

26 Dec 2020

Using the VERT wearable device to monitor jumping loads in elite volleyball athletes

PONE-D-20-21022R2

Dear Dr. Scott,

We’re pleased to inform you that your manuscript has been judged scientifically suitable for publication and will be formally accepted for publication once it meets all outstanding technical requirements.

Kind regards,

José M. Muyor

Academic Editor

PLOS ONE

---

## [Editor Report · Acceptance letter]

2 Jan 2021

PONE-D-20-21022R2 

Using the VERT wearable device to monitor jumping loads in elite volleyball athletes 

Dear Dr. Scott:

I'm pleased to inform you that your manuscript has been deemed suitable for publication in PLOS ONE. Congratulations! Your manuscript is now with our production department. 

Kind regards, 

on behalf of

Dr. José M. Muyor 

Academic Editor

PLOS ONE